# Impact of Carpal Tunnel Syndrome Surgery on Early Diagnosis and Treatment of Transthyretin Cardiac Amyloidosis

**DOI:** 10.3390/medicina59020335

**Published:** 2023-02-10

**Authors:** Inesa Kuznecova, Gerda Mierkyte, Dainius Janciauskas, Donatas Vajauskas, Antanas Jankauskas, Loreta Pilipaityte, Rytis Rimdeika, Vytautas Tamaliunas, Egle Ereminiene

**Affiliations:** 1Department of Cardiology, Medical Academy, Lithuanian University of Health Sciences, Kaunas Region Society of Cardiologists, LT-44307 Kaunas, Lithuania; 2Faculty of Medicine, Medical Academy, Lithuanian University of Health Sciences, LT-44307 Kaunas, Lithuania; 3Department of Pathological Anatomy, Medical Academy, Lithuanian University of Health Sciences, LT-44307 Kaunas, Lithuania; 4Department of Radiology, Medical Academy, Lithuanian University of Health Sciences, LT-44307 Kaunas, Lithuania; 5Department of Plastic and Reconstructive Surgery, Medical Academy, Lithuanian University of Health Sciences, LT-44307 Kaunas, Lithuania; 6Laboratory of Clinical Cardiology, Institute of Cardiology; Kaunas Region Society of Cardiologists, LT-44307 Kaunas, Lithuania

**Keywords:** cardiac amyloidosis, transthyretin, carpal tunnel syndrome, case report, rare diseases

## Abstract

*Background and Objectives*: Cardiac amyloidosis is an infiltrative, progressive, and restrictive cardiomyopathy that leads to heart failure, reduces life quality, and causes death. This is a multisystem disorder caused by mutations of the transthyretin protein and is associated not only with cardiac diseases or carpal tunnel syndrome but also with nerve, liver, lung, gastrointestinal tract, kidney, or eye pathologies. Carpal tunnel syndrome is an early red-flag symptom of transthyretin (TTR) cardiac amyloidosis; therefore, screening for unsuspected cardiac amyloidosis can be performed through histological testing of flexor retinaculum specimens gathered during carpal tunnel release surgery. Our case highlights that early detection and accurate diagnosis of a disease are important factors for improving clinical outcomes in patients with TTR amyloidosis. *Case Summary*: We report the case of a 71-year-old man who presented with bilateral carpal tunnel syndrome. Amyloid deposits were detected after carpal tunnel release surgery through histological testing of the synovial tissue. The patient was sent for a cardiological evaluation. Physical examination, laboratory tests, and the ECG revealed no significant changes. The diagnosis of amyloidosis was confirmed with multimodality imaging in the early stage, which helped to start specific medicamental treatment with the transthyretin stabilizer tafamidis. *Conclusions*: Our objective is to highlight the early recognition and specific medicamental treatment of cardiac amyloidosis for better patient prognosis and outcomes.

## 1. Introduction

Cardiac amyloidosis (CA) is an infiltrative and restrictive cardiomyopathy that leads to heart failure, reduces life quality, and causes death. The disease has two subtypes, one of which is mutant or wild-type transthyretin amyloidosis (ATTRwt), which is caused by a representative amyloidogenic protein in humans, produced predominantly by the liver [1], that transports thyroid hormone and retinol [2]. Normally, transthyretin (TTR) is a tetramer, a stable form, found in the bloodstream. However, with aging or a destabilizing mutation, it becomes unstable, develops into misfolded monomers or oligomers, and ultimately deposits as amyloid fibrils [3] in different organs and tissues, causing morphological and functional alterations. Deposition in the myocardium is the most important prognostic factor that represents the severity of the disease [4]. Additionally, a significant proportion of patients with ATTR have deposition in soft tissue structures leading to spinal stenosis, biceps tendon rupture, and carpal tunnel syndrome (CTS) [5]. CTS is the most common upper-extremity nerve compression syndrome. It is an entrapment of the median nerve in the wrist joint associated with reduced sensation, dexterity, and function of the nerve [6]. Studies have shown that in patients undergoing carpal tunnel release surgery, 10–15% have a positive biopsy for amyloid and 2–10% have cardiac involvement [4,7,8,9]. CTS is frequently bilateral and typically manifests in patients with TTR amyloidosis, most of the time 5–10 years prior to cardiac diagnosis, usually at the age of 50–70, often requiring carpal tunnel release surgery [9,10,11,12]. The study data suggest that compression syndromes, such as CTS and spinal stenosis, may be predictive of the future of systemic amyloid diseases and could help with early treatment and improvement in the clinical outcomes of the patients with ATTR [13]. The treatment of patients with TTR amyloidosis is frequently delayed, owing to low disease awareness [14,15] or incorrect diagnosis of the pathology, such as hypertrophic cardiomyopathy or hypertensive heart disease [16]. Thus, many patients remain undiagnosed for a long period of time [3] because of the limited specificity of echocardiography and other tests, the patients‘ age, and comorbidities, and it is almost always associated with a poor prognosis [2]. In this report, we describe the clinical case of a 71-year-old man who underwent carpal tunnel surgery. A positive biopsy for transthyretin amyloid and the noninvasive multimodality imaging provided an early ATTRwt diagnosis, and specific treatment was started in time. Although testing patients may provide an early diagnosis of ATTR, the disease is rarely diagnosed in clinical practice, and such patients are not routinely tested for amyloidosis. In this clinical case, we want to emphasize that patients who undergo surgery for bilateral carpal tunnel syndrome have a much higher risk of developing cardiac amyloidosis; therefore, they must be continuously monitored and treated if necessary.

## 2. Case Report

A 71-year-old man was diagnosed with idiopathic bilateral carpal tunnel syndrome and admitted to the hospital of the Lithuanian University of Health Sciences, Department of Plastic and Reconstructive Surgery, for surgical treatment. He has been treated for arterial hypertension and hypercholesterolemia with perindopril/indapamid/amlodipine (10/2.5/5 mg/d) and atorvastatinum, dose-selecting depending on cholesterol level. No other cardiac diseases had been diagnosed before the surgery.

The samples taken from the synovial tissue on the flexor tendons were analyzed, and amyloid deposits were detected by staining the samples in Congo Red and Sirius Red; the TTR type of amyloid was identified by immunohistochemical staining with anti-TTR (Figure 1). The patient was sent for a cardiological evaluation. The physical examination revealed no significant changes. The patients had I functional class heart insufficiency, as classified by the New York Heart Association (NYHA). The blood pressure (BP) was 140/82 mmHg and the heart rate was 68 beats/min. A 6 min distance test estimated good physical capacity (460 m) with blood oxygen saturation (SpO2) up to 96%.

The results of the laboratory tests were as follows: the concentration of natriuretic peptide (BNP) was 64 ng/L (normal ranges < 26 ng/L), the troponin I level was 0.04 $\frac{\mu g}{l}\;$, normal renal function. The urine κ/λ free light chains ratio was increased to 11, while this ratio was normal (0.8), and no monoclone protein was detected in blood serum. ECG revealed sinus rhythm with low voltage in standard leads and poor R wave progression in left precordial leads without conduction disturbances and arrhythmias (Figure 2).

A two-dimensional echocardiographic examination (Figure 3) revealed left ventricle (LV) concentric hypertrophy without dilation, with preserved LV ejection fraction (LVEF of 60%) and pseudonormal diastolic dysfunction (E/E’—10, left atrial (LA) dilatation of 38 mL/m^2^), and reduced deformation LA parameters (left atrial reservoir strain of 22.3%). The global longitudinal LV strain was slightly reduced (−17%), with maintained strain values in the apical segments. The right-sided heart chambers were not dilated.

Cardiac magnetic resonance imaging (MRI) (Figure 4) found signs of infiltrative myocardial damage. The LV volumes were not dilated, with an interventricular septum thickened up to 20 mm and a maintained LVEF of 67%. Native T1 values were elevated to 1321 ms. LGE analysis showed a linear and focal diffuse delayed accumulation of gadolinium, with prevalent subendocardial late gadolinium enhancement (LGE) accumulation in the basal and middle LV segments.

To confirm the presence and type of amyloidosis, technetium-99m pyrophosphate scintigraphy and single photon emission computed tomography (SPECT) were performed (Figure 5). The results obtained using the qualitative assessment scale corresponded to degree 3 (Perugini scale). A genetic test showed no mutations in the TTR gene. Based on these findings, wild-type transthyretin cardiac amyloidosis was diagnosed.

The patient was under active follow-up for ten months. During this period, the heart failure symptoms started to progress, and the BNP increased to 121.0 ng/l. According to the 2021 ESC guidelines for diagnosis and management of heart failure, in order to stop the progression of the disease, the specific treatment with tafamidis 61 mg/d was started along with the optimal medicamental arterial hypertension and dyslipidemia treatment, with close follow-up every three months.

## 3. Discussion

TTR amyloidosis is a life-threatening, progressive, and infiltrative disease that can often be underdiagnosed and overlooked as a common cause of heart failure (HF). Early detection and accurate diagnosis of a disease are important factors for improving clinical outcomes in patients with TTR amyloidosis.

CTS is an early red-flag symptom of TTR amyloidosis, which appears in 10–15% [4,7,8,9] of 50–70-year-old patients with an amyloid-positive biopsy of the tenosynovium [8]. CTS classically presents bilaterally, and it appears years before cardiac and multisystem involvement [8]. Studies have shown that four factors, such as male sex, older age, cardiac involvement, and wild-type disease, are associated with CTS in TTR amyloidosis [4]. The biological link among TTR amyloidosis, CTS, heart failure, and other cardiovascular disorders is the systemic accumulation of amyloid over time [7]. TTR amyloidosis is a systemic disease that is associated not only with cardiac diseases, or CTS, but also with nerve, liver, lung, gastrointestinal tract, kidney, or eye pathologies [3]. Clinical signs, such as heart failure, fatal arrhythmia, severe orthostatic hypotension, respiratory muscle paralysis, kidney failure, nephrotic syndrome, protein-losing gastroenteropathy, and severe glaucoma, occur in the later stages of the disease [3]. Severe peripheral neuropathy, cardiac, renal, and respiratory disorders can often lead to death when TTR amyloidosis is left undiagnosed and untreated. For this reason, the survival rate of this disease remains approximately 3–10 years [3,17,18].

Cardiac amyloidosis can be diagnosed based on a family history of cardiomyopathy, gastrointestinal disorders, unexplained weight loss, CTS, neuropathy, renal or ocular impairment, clinical examination, blood tests, and instrumental investigations [19]. Consequently, screening for unsuspected cardiac amyloidosis can be performed through histological testing of flexor retinaculum specimens gathered during carpal tunnel release surgery [13]. Perioperative biopsies of the tenosynovium allow for the detection of TTR amyloidosis at an early stage before CA develops, extending the patient’s survival time by several years [17]. A biopsy of the tendon tissue must be taken during the surgery and sent to the pathological anatomy clinic in standard formalin for Congo Red or Sirius Red staining [8]. Polarized light microscopy does not provide any information about the amyloid type; therefore, mass spectrometry must be ordered to define the subtype if Congo Red staining is positive [8]. If TTR amyloid is detected, patients should be examined with further testing by electrocardiography, echocardiography, cardiac-specific biomarkers, genetic tests, and nuclear imaging to diagnose cardiac amyloidosis and determine specific treatment [13].

The goal of the TTR amyloidosis treatment is to provide medical support and, if possible, to stop or delay amyloid deposition through the use of specific treatments. Recently, pharmacological drugs for patients with ATTR were approved [2]. Historically, symptomatic management of TTR amyloidosis was the only option. Since the 1990s, the way to eliminate the main source of precursor TTR has been liver transplantation alone or in combination with heart transplantation [20] in the hereditary type of the disease. Until specific CA treatment was approved, the median life expectancy was 3.5 years after CA diagnosis [21]. Nowadays, early diagnosis of amyloidosis gives an opportunity to prolong life expectancy through various therapeutic modalities, such as transthyretin stabilizers (e.g., diflunisal or tafamidis, TTR gene silencers) or amyloid fibrils degraders (ursodeoxycholic acid or epigallocatechin 3-gallate found in green tea extracts) [22,23]. Tafamidis, a nonsteroidal anti-inflammatory benzoxazole derivative, inhibits TTR dissociation of tetramers into monomers and suppresses deposition in tissues [24]. Studies observed a consistent benefit from therapy with tafamidis, such as reduced cardiovascular-related hospitalizations, especially in patients with NYHA classes I/II [18]. Thus, decreasing mortality and significantly reducing the decline in functional capacity and quality of life [24]. Diflunisal belongs to the same class of medicines as tafamidis, but this pharmacological drug is not tolerated by many patients with TTR amyloidosis. Data suggest that diflunisal worsens volume overload and renal dysfunction but reduces the progression of neuropathy and preserves the quality of life [5].

A low-risk biopsy of the tenosynovium is extremely important for the early diagnosis of cardiac amyloidosis and the initiation of specific treatment for patients diagnosed with bilateral CTS. In 2018, Sperry et al. conducted a prospective, cross-sectional, multidisciplinary study of men aged over 50 and women aged over 60 undergoing carpal tunnel release surgery. In this study, the authors found that 10.2% of patients had an amyloid-positive biopsy of the tenosynovium. Those specimens were confirmed by mass spectrometry, echocardiography, cardiac-specific biomarkers, free light chain immunofixation electrophoresis, and nuclear imaging to define the presence of cardiac amyloidosis. The authors proved that a tenosynovial biopsy taken during carpal tunnel release surgery can lead to an early diagnosis of amyloidosis and may prevent progressive heart failure [9]. Another observational study was based on Danish nationwide registries from 1996 to 2012 of patients who had carpal tunnel release surgery. The aim of the study was to examine their risk of amyloidosis, heart failure, and other cardiovascular diseases. The authors investigated the possibility that CTS, in some cases, may be an early warning sign of amyloidosis and can be a marker of HF and other adverse cardiovascular outcomes [7]. Other older studies have also shown amyloid deposition in 12–35% of patients with idiopathic carpal tunnel syndrome, so it was concluded that early recognition is important to initiate appropriate disease-modifying therapy and prevent systemic amyloidosis [25,26].

Patients with cardiac amyloidosis have a poor prognosis if the disease goes undiagnosed and untreated [8,9], because the patient’s condition deteriorates rapidly due to systemic amyloid deposition and advancing organ dysfunction [15]. According to the international survey by McCausland et al. (2018), amyloidosis was misdiagnosed in more than half of patients with hereditary ATTR and approximately one third of patients with ATTRwt and immunoglobulin light chain amyloidosis (AL). It has been noticed that most of the patients visited three or more different physicians before receiving a correct diagnosis [27]. In most cases, CA is diagnosed in the later stages, following the manifestation of severe cardiac symptoms [2,28]. This disease results in progressive ventricular stiffness, wall thickening, and diastolic filling abnormalities, which typically present as restrictive physiology and heart failure with preserved ejection fraction (HFpEF) [29].

In our case, the patient had arterial hypertension and hypercholesterolemia with minimal heart failure symptoms. Carpal tunnel syndrome surgical treatment and histological tenosynovium analysis showed amyloidosis, which was confirmed with multimodality imaging. ATTRwt was diagnosed in its early stages thanks to genetic testing, which aided in the early initiation of specific medicinal treatment with the transthyretin stabilizer tafamidis. We strongly believe that the histological tenosynovium evaluation for amyloid deposition in patients with bilateral CTS undergoing surgical treatment, especially in older male patients, can aid in the earlier recognition of this rare multysistemic disease and improve clinical outcomes with early specific medical treatment.

## 4. Conclusions

ATTRwt is a late-onset disease, and symptoms manifest mostly in elderly patients with comorbidities that blur the correct diagnosis. Patients with bilateral idiopathic CTS should be investigated with a tenosynovial biopsy, as an early histological diagnosis results in a better patient prognosis and outcome.

## Figures and Tables

**Figure 1 medicina-59-00335-f001:**
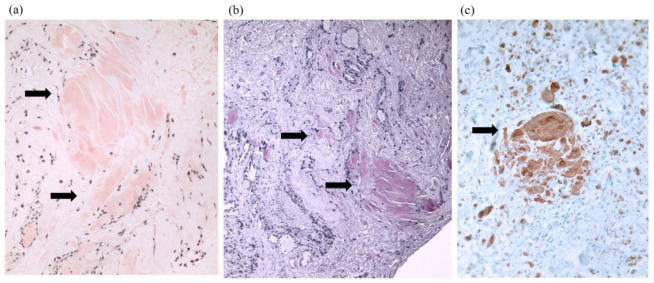
Congo Red stain (**a**), Sirius Red stain (**b**), and imunohistochemistry with anti-TTR (**c**). Arrows point amyloid (transthyretin) deposition in the flexor tenosynovium.

**Figure 2 medicina-59-00335-f002:**
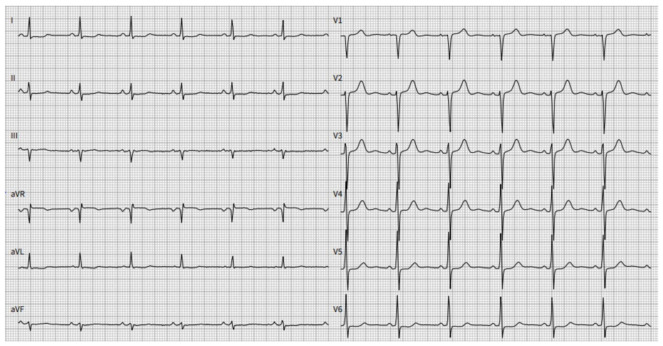
A twelve-lead electrocardiogram (ECG) demonstrated sinus rhythm with a normal rate. Poor R wave progression and low-voltage QRS were observed in standard leads. PR interval 139 ms, Qtc interval 424 ms.

**Figure 3 medicina-59-00335-f003:**
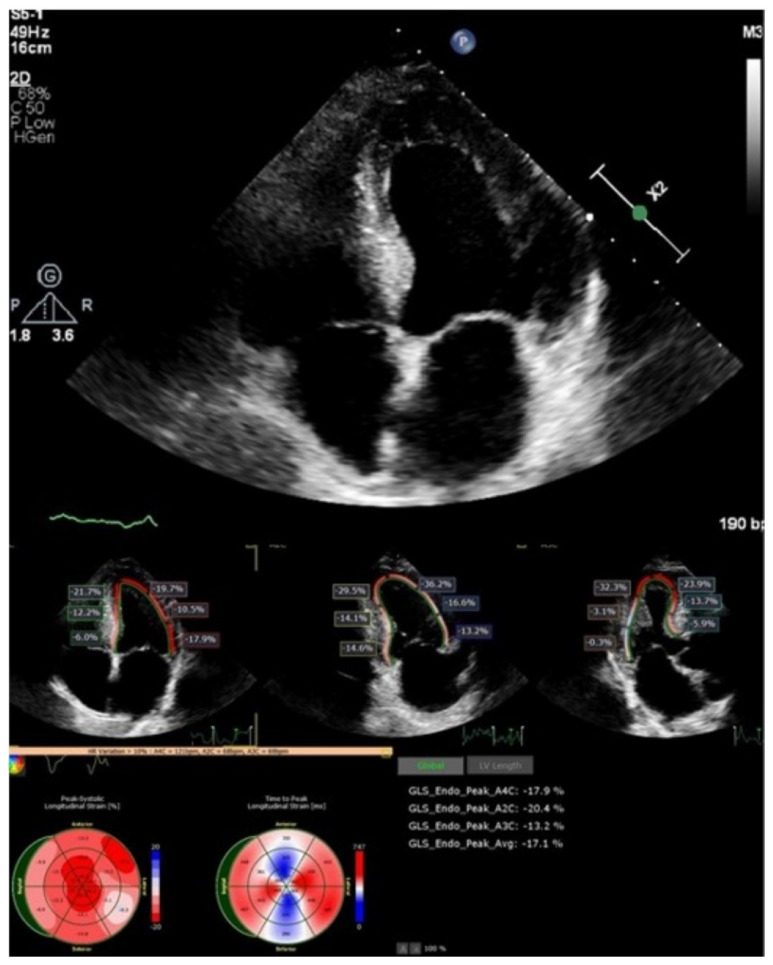
Two-dimensional echocardiography and speckle tracking: LV hypertrophy with maintained strain in apical LV segments.

**Figure 4 medicina-59-00335-f004:**
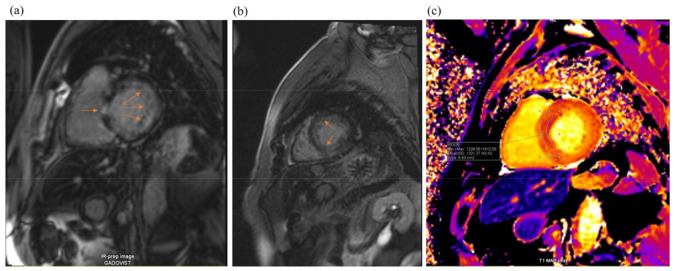
Cardiac magnetic resonance imaging: LV hypertrophy with diffuse subendocardial late gadolinium enhancement (arrowheads, pictures (**a**,**b**)) and some zones of focal late gadolinium enhancement (arrow, picture (**c**)). T1 mapping shows markedly increased values (1321 ms).

**Figure 5 medicina-59-00335-f005:**
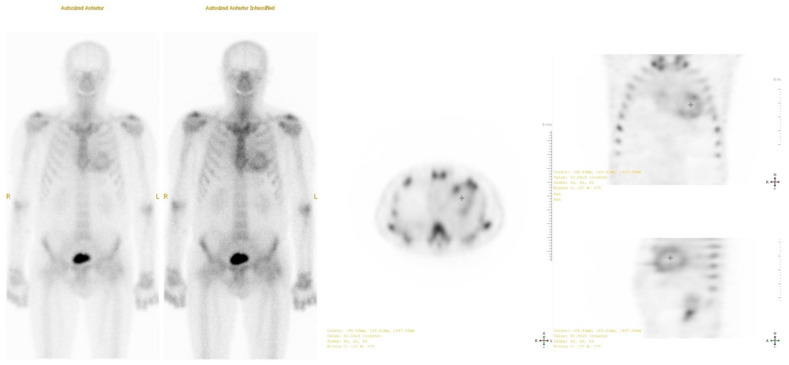
Technetium-99 m pyrophosphate single photon emission computed tomography (SPECT): positive uptake of radiotracer in the heart 1 h after injection (Grade 3).

## Data Availability

Not applicable.

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
