# Peer review of "Impact of Carpal Tunnel Syndrome Surgery on Early Diagnosis and Treatment of Transthyretin Cardiac Amyloidosis"

_medicina, 2023, doi:10.3390/medicina59020335_

Round 1
Reviewer 1 Report
Impact of carpal tunnel syndrome surgery on early diagnosis and treatment of transthyretin cardiac amyloidosis
Inesa Kuznecova et al.
You do not explain all abbreviations even if common (ATTR rows 20, 24 etc; LGE rows 106, 107; LV and so on)... please see https://www.mdpi.com/journal/information/instructions > Manuscript Preparation > Acronyms/Abbreviations/Initialisms
You use both ATTR amyloidosis and TTR amyloidosis (rows 137 and 139) while you can use TTR amyloidosis or ATTR (the A stands for amyloidosis) as you explained and used in row 40 [transthyretin (TTR) amyloidosis]...
ATTR can be inherited (ATTRm) or acquired (ATTRwt)... please review abbreviations...
Figure 3 writing is not clear... can you provide a better resolution?
Can you describe the details of the genetic test (row 116)
Please check the manuscript for moderate English changes (there are some long phrases that you should shorten: rows 213-217 etc)
Insist on the particularity of the case from Introduction to the Conclusion (why publishing the article would be useful): age, comorbidities, prognosis. Can you comment more on the evolution? (you can see https://www.care-statement.org/checklist)
Please describe the Informed Consent and include no/date of approval... please see https://www.mdpi.com/journal/medicina/instructions
Please review References (remove "[Internet]" after the name of Journals etc.)... please see https://www.mdpi.com/journal/information/instructions > Back Matter > References and https://www.mdpi.com/authors/references
Author Response
Impact of carpal tunnel syndrome surgery on early diagnosis and treatment of transthyretin cardiac amyloidosis
Inesa Kuznecova et al. Manuscript ID: medicina-2183643.
First of all, thank you very much for your comments and detailed explanation of what should be corrected. I have fixed everything and I am sending my response.
Point 1. You do not explain all abbreviations even if common (ATTR rows 20, 24 etc; LGE rows 106, 107; LV and so on)... please see https://www.mdpi.com/journal/information/instructions > Manuscript Preparation > Acronyms/Abbreviations/Initialisms
Response 1. I have explained.
Point 2. You use both ATTR amyloidosis and TTR amyloidosis (rows 137 and 139) while you can use TTR amyloidosis or ATTR (the A stands for amyloidosis) as you explained and used in row 40 [transthyretin (TTR) amyloidosis]...
Response 2. Now I use only TTR amyloidosis or ATTR without word amyloidosis.
Point 3. ATTR can be inherited (ATTRm) or acquired (ATTRwt)... please review abbreviations...
Response 3. Now I use these abbreviations in my work.
Point 4. Figure 3 writing is not clear... can you provide a better resolution?
Response 4. Unfortunately, no.. We do not have other picture of this measurement.
Point 5. Can you describe the details of the genetic test (row 116)
Response 5. We only investigated the mutation is present or not. Genetic test showed no mutations in the TTR gene, I do not have more details.
Point 6. Please check the manuscript for moderate English changes (there are some long phrases that you should shorten: rows 213-217 etc)
Response 6. I have moderated it.
Point 7. Insist on the particularity of the case from Introduction to the Conclusion (why publishing the article would be useful): age, comorbidities, prognosis. Can you comment more on the evolution? (you can see https://www.care-statement.org/checklist)
Response 7. I didn't fully understand the comment but I tried to emphasize the most important idea in the Introduction. This answers the question of why this clinical case is important and needed. We have used care statement guidelines now and corrected by these.
Point 8. Please describe the Informed Consent and include no/date of approval... please see https://www.mdpi.com/journal/medicina/instructions
Response 8. I added the sentence - "written informed consent has been obtained from the patient to publish this paper".
I sent the document for MDPI Medicina appointed editor exact after I had the form 13/01/23.
Point 9. Please review References (remove "[Internet]" after the name of Journals etc.)... please see https://www.mdpi.com/journal/information/instructions > Back Matter > References and https://www.mdpi.com/authors/references
Response 9. I corrected as recommended.
Reviewer 2 Report
Dear authors,
Thanks for the opportunity of reviewing your manuscript. This work presents a case report that highlight the need of early diagnose of cardiac amyloidosis, considering the carpal tunnel syndrome. However, it is an interesting research topic some important issues need to be addressed before being considered for pubication. I hope my comment help the authors to improve the quality of their manuscript.
# General comments.
The major issue in this study is the lack of a proper literature background, to justify why this case report is need, and what new is added to the literature. This has to be addressed. Another major issue is that authors should have followed the CARE checklist to write their manuscript. For example, a discussion section have to be included. This statement has to be followed for the entire manuscript. Also, some minor problems are the need to clarify what ATTR means or to provide more information about carpal tunnel syndrome.
Abstract:
# Comment 1: ATTR needs to be clarify in order to help the reader.
# Comment 2: Conclusion can not repeat the same information provided in the lines 23 and 24 of the abstract. It has to provide a conclusion based on the case report.
Introduction:
# Comment 1: Please, same as in the abstract the full name of ATTR has to be provided.
# Comment 2: I congrats the authors for the definitions and characteristics of the cardiac disease, but carpal tunnel syndrome needs to be explained in depth for a general view of the complete case.
# Comment 3: A major issue of your study is the lack of background research. Authors have to provide a concise state of the art to make a proper justification of their work. Why is this case report needed? What does this case report add new to the current literature? Please, authors have to addressed this major issue.
Case report
# Comment 1: Did the authors follow the CARE guidelines? Please see: https://www.care-statement.org/checklist
# Comment 2: Please, if possible, improve the quality of Figure 5.
Discussion:
# Comment 1: Please, authors should follow the CARE guidelines, and have to include a discussion section.
Author Response
Impact of carpal tunnel syndrome surgery on early diagnosis and treatment of transthyretin cardiac amyloidosis
Inesa Kuznecova et al. Manuscript ID: medicina-2183643.
First of all, thank you very much for your comments and detailed explanation of what should be corrected. I have fixed everything and I am sending my response.
Point 1: ATTR needs to be clarify in order to help the reader.
Response 1: I have explained all non clarify acronyms now.
Point 2: Abstract conclusion can not repeat the same information provided in the lines 23 and 24 of the abstract. It has to provide a conclusion based on the case report.
Response 2: Thank you, I wrote a conclusion based on a clinical case.
Point 3: Please, same as in the abstract the full name of ATTR has to be provided.
Response 3: Thank you, already did.
Point 4: I congrats the authors for the definitions and characteristics of the cardiac disease, but carpal tunnel syndrome needs to be explained in depth for a general view of the complete case.
Response 4: I added an explanation of carpal tunnel syndrome.
Point 5: A major issue of your study is the lack of background research. Authors have to provide a concise state of the art to make a proper justification of their work. Why is this case report needed? What does this case report add new to the current literature? Please, authors have to addressed this major issue.
Response 5. Thank you. In the introduction, I tried to emphasize the most important idea, this answers the question of why this clinical case is important and needed.
Point 6: Did the authors follow the CARE guidelines? Please see: https://www.care-statement.org/checklist
Response 6: I have corrected our article following the guidelines
Point 7. Please, if possible, improve the quality of Figure 5.
Response 7: Unfortunately, we can not. We do not have other picture of this measurement.
Point 8: Please, authors should follow the CARE guidelines, and have to include a discussion section.
Response 8: We have expanded the discussion section and added the missing parts following the CARE guidelines
Round 2
Reviewer 2 Report
Dear authors,
Thank you for the effort in addressing all my queries. The manuscript has improved the quality, and I have no more comments to add.